# Prognostic Significance of Interim Response Evaluation during Definitive Chemoradiotherapy for Locally Advanced Esophageal Squamous Cell Carcinoma

**DOI:** 10.3390/cancers13061255

**Published:** 2021-03-12

**Authors:** Jun Gi Yeom, Jie-Hyun Kim, Jun Won Kim, Yeona Cho, Ik Jae Lee, Chang Geol Lee, Jaeyoung Chun, Young Hoon Youn, Hyojin Park

**Affiliations:** 1Department of Internal Medicine, Gangnam Severance Hospital, Yonsei University College of Medicine, Seoul 06273, Korea; duawnsrl24@gmail.com (J.G.Y.); CHUNJMD@yuhs.ac (J.C.); DRYOUN@yuhs.ac (Y.H.Y.); HJPARK21@yuhs.ac (H.P.); 2Eastern-Seoul Branch, Korea Association of Health Promotion, Seoul 07572, Korea; 3Departments of Radiation Oncology, Gangnam Severance Hospital, Yonsei University College of Medicine, Seoul 06273, Korea; IAMYONA@yuhs.ac (Y.C.); IKJAE412@yuhs.ac (I.J.L.); 4Department of Radiation Oncology, Severance Hospital, Yonsei University College of Medicine, Seoul 03722, Korea; CGLEE1023@yuhs.ac

**Keywords:** esophageal cancer, definitive CRT, adaptive radiotherapy, adaptive CT, interim response

## Abstract

**Simple Summary:**

We investigated the clinical significance of interim response evaluation during definitive chemoradiotherapy in locally advanced esophageal squamous cell carcinoma. Interim response was evaluated using adaptive CT images including primary esophageal lesion and lymph node. The reduction rate of tumor area or diameter was measured. Interim response correlated with complete response and survival rates. That is, the evaluation of tumor burden reduction during treatment may help predict patient prognosis.

**Abstract:**

The study aimed to investigate the clinical significance of interim response evaluation during definitive chemoradiotherapy (dCRT) in predicting overall treatment response and survival of patients with locally advanced esophageal squamous cell carcinoma (LAESCC). We reviewed 194 consecutive patients treated with dCRT for biopsy-confirmed LAESCC. A total of 51 patients met the inclusion criteria. Interim response was assessed by defining a region of interest in initial and adaptive computed tomography (CT) images and subsequently examined against the overall treatment response assessed three months after dCRT, treatment failure pattern, overall survival (OS), and progression-free survival (PFS) estimates. Reductions in both the area and maximal diameter of the primary lesion (*p* < 0.001; *p* < 0.001, respectively) and those of the metastatic lymph nodes (LN) (*p =* 0.002; *p* < 0.001, respectively) in interim analysis were significantly higher among patients who achieved complete response (CR) than among those who did not. OS was significantly longer among patients who showed ≥30% interim reduction in the area and maximal diameter of the primary lesion and among those who showed such reduction in both the primary lesion and LN. PFS was significantly longer in the patients with ≥30% interim reduction in the area of the primary lesion. In addition, the proportion of cases with locoregional failure began decreasing at interim response of 20% or higher, while the proportion of cases with outfield failure followed the opposite pattern, increasing at interim response of 20% or higher. Among patients treated with dCRT for LAESCC, interim response assessed using adaptive CT images correlated with overall CR and OS rates. The evaluation of tumor burden reduction during dCRT may help predict patient prognosis.

## 1. Introduction

Esophageal cancer is the eighth most common cancer worldwide; it is associated with poor prognosis, which makes it also the sixth leading cause of cancer-related deaths [1]. Squamous cell carcinoma is the most prevalent histological subtype of esophageal cancer, while the incidence of adenocarcinoma is increasing in Western countries [2,3]. A staging system based on tumor-node-metastasis subclassification is available. However, esophageal cancers are often categorized according to the indicated treatments into early-stage superficial cancers, locally-advanced cancers with, or without, regional lymph node (LN) metastases in the absence of distant metastases, and cancers with distant metastases [4]. Patients with locally advanced esophageal cancer are a heterogeneous group that can be further divided into those with potentially resectable (T2 up to T4a) and unresectable (T4b) primary disease, poor surgical candidates, and those who decline surgery [5].

Surgery had been regarded as a standard treatment for patients with resectable esophageal cancer. However, patient outcomes remain unsatisfactory with median survival rarely exceeding 18 months [6]. Moreover, most patients are diagnosed when the disease has reached advanced or unresectable stages. Standard therapy for locally advanced esophageal squamous cell carcinoma (LAESCC) includes neoadjuvant chemotherapy or chemoradiotherapy (CRT) followed by surgery or definitive CRT (dCRT) [7]. Patients undergoing dCRT may experience local recurrence and require salvage surgery, while two-year survival rates have been reported as 31–40% [8,9,10].

Adaptive radiation therapy (ART) is a mid-treatment process where the initial radiotherapy (RT) plan is modified to account for the anatomical changes in the tumor or organs at risk, aiming to deliver an accurate dose of radiation to the target while minimizing the risk of toxicity [11]. At our institution, ART consists of a second set of computed tomography (CT) scanning simulation and RT planning, usually three to five weeks into dCRT to accommodate anatomical changes due to tumor shrinkage. These may provide additional information regarding early (interim) treatment response. In contrast, conventional treatment response evaluation is performed one to three months after treatment completion, based on a comparison of pre- and post-treatment radiologic studies.

The present study aimed to evaluate the role of interim response evaluation during dCRT for LAESCC patients in predicting overall treatment response and survival rates. The differences in treatment failure patterns based on the interim response were also investigated.

## 2. Materials and Methods

### 2.1. Patient Identification

Patients who underwent dCRT for the treatment of locally advanced (stages II–IV) esophageal cancer at the Gangnam Severance Hospital and Severance Hospital between January 2005 and December 2018 were screened for eligibility. Only biopsy-confirmed squamous cell carcinoma cases with a measurable primary esophageal lesion on a baseline CT scan were included in the analysis. Patients were excluded from the present study if they did not undergo a CT scanning simulation for ART during dCRT or if they underwent esophageal stent insertion before or during dCRT. In general, patients underwent ART if the initial CT showed the gross tumor deviating the normal anatomy of the esophagus (usually ≥T3) and/or patients complained of symptomatic dysphagia (≥Grade 2, CTCAE 4.0) during the initial physical examination. Among 194 patients with LAEC treated with dCRT, patients with non-squamous cell carcinoma histology, an unmeasurable primary esophageal lesion in CT, esophageal stent insertion, and no adaptive CT images were excluded. Patients were assessed by endoscopic ultrasonography, chest and abdominal CT scans, and ^18^F-fluorodeoxyglucose positron emission tomography or positron emission tomography-CT (PET-CT) scans. The eighth edition of the American Joint Commission on Cancer stating system was used [4].

### 2.2. Treatment Regimens

All patients underwent definitive concurrent CRT. RT was performed with three-dimensional conformal RT or intensity-modulated RT using conventional fractionation schedules (5 days/week, 1.8–2.0 Gy/daily fraction) and the cone-down technique at the time of ART. Gross tumor volume means primary tumor and metastatic regional LNs on images. Clinical target volume was determined as gross tumor volume plus 3 cm craniocaudally, 1 cm laterally, and 3 cm into the gastric mucosa in case of gastroesophageal junction tumors. A uniform 0.5-cm expansion from clinical target volume was added to define planning target volume. The regimen of concurrent chemotherapy was mainly cisplatin and 5-fluorouracil. The dose and schedule was 75 mg/m^2^ on day 1 for cisplatin and 1000 mg/m^2^ on days 1–4 for 5-fluorouracil, every 4 weeks in most LAESCC patients.

### 2.3. Defining Regions of Interest and Interim Analysis at ART

Contrast-enhanced initial and adaptive CT images were loaded into the Picture Archiving and Communication System software (Centricity^TM^ PACS, GE Healthcare, Slough, United Kingdom) for the measurement of lesions. Initial CT scanning included diagnostic CT images taken at baseline and simulation CT images acquired for initial RT planning. An experienced radiation oncologist selected axial CT slices containing the primary esophageal lesion and the regional metastatic LNs with the largest diameters. In these axial CT images, regions of interest (ROIs) were manually delineated along the boundaries of tumor lesions. The area of ROI was measured including the necrotic area within the tumor, while carefully excluding adjacent fluid or air volumes or normal esophagus. The maximal diameter of esophageal cancer lesion and diameter of LN were also measured. Changes in these measurements between initial CT and adaptive CT images were calculated. An example of an ROI measuring is provided in Figure 1.

### 2.4. Assessment of Overall Treatment Response

The overall treatment response was assessed on endoscopy and chest CT, abdominal pelvic CT, and PET-CT scans 3 months after the end of dCRT, following the Response Evaluation Criteria for Solid Tumors (RECIST) version 1.1 [12]. A complete response (CR) was defined as tumor residue not visible on endoscopy, histologic examination, chest CT and PET scan. CR was decided considering all examinations of endoscopy, chest CT, and PET. After dCRT for esophageal tumors, a thickening of the esophageal wall can remain in chest CT. Thus, chest CT was used mainly to evaluate CR for LN not primary lesion. To evaluate the CR for primary esophageal lesion was mainly endoscopic biopsy and PET. Gross tumor disappearance on endoscopy, no tumor cell on biopsy, and normalization of tumor FDG activity on PET were defined CR for the esophageal lesion. Treatment failure was defined as disease recurrence after achieving CR or incomplete remission after dCRT. Treatment failure was sub-classified into locoregional and outfield failure. Locoregional failure is a treatment failure in the esophagus or regional LN within the RT field, whereas outfield failure is a treatment failure in other areas.

### 2.5. Statistical Analyses

The index date was day of LAESCC diagnosis. All enrolled patients were followed-up until their last visit, death, or October 2019, whichever occurred first. The chi-square or Fisher’s exact tests and Student *t*-tests were used to analyze statistical correlations between categorical, and non-categorical variables. Overall survival (OS) was measured from the date of diagnosis to the date of death or the date of the last follow-up visit. Progression-free survival (PFS) was measured from the date of treatment initiation until disease progression or worsening. OS and PFS were analyzed by the Kaplan–Meier method. Multivariate analysis was performed by logistic regression and Cox regression model. A *p*-value of < 0.05 was considered significance. The IBM SPSS Statistics for Windows, Version 25.0. (Armonk, NY, USA: IBM Corp) and SAS version 9.4 (SAS Institute, Cary, NC, USA) was used for statistical analyses.

### 2.6. Ethical Considerations

The present study was performed in accordance with the Helsinki Declaration and approved by the institutional review board of Gangnam Severance Hospital (3-2018-0338). All data were de-identified and anonymized.

## 3. Results

### 3.1. Baseline Characteristics of Patients

Among 194 screened patients, 51 met the inclusion criteria, and their treatment data were analyzed. Baseline characteristics of the patients are shown in Table 1. The mean age of the patients was 65.6 ± 9.1 years, 43 (84.3%) were male, clinical stage III was the most frequent (31, 60.8%), and most patients (45, 88.2%) received chemotherapy with the 5-fluorouracil plus cisplatin regimen. The median total radiation dose was 6300 (4200–7200) cGy. The median follow-up duration was 16.6 (10.2–41.3) months.

### 3.2. Interim Analysis

The mean time interval between the initial and adaptive CT scan was approximately 1 month (30.6 ± 7.3 days). The interim response evaluation revealed mean reduction of 33.7 ± 24.0% in the area and 32.0 ± 23.4% in the diameter of the primary esophageal lesion. In the case of LN, 27.2 ± 25.1% reduction in the area and 16.6 ± 19.2% reduction in the diameter were observed. When the interim responses of both primary tumor and LNs were combined, the mean reductions observed were 32.4 ± 22.4% and 25.8 ± 22.6% in the area and diameter, respectively (Table 2).

### 3.3. Overall Treatment Response and Pattern of Failure

Of 51 patients, 26 (51%) showed CR 3 months after dCRT and 9 (35.6%) showed recurrence during the follow-up period. Treatment failure, including incomplete remission and recurrence after achieving CR, were reported in 34 (66.7%) patients. Among 34 failure cases, 16 (47.1%) were locoregional (esophageal recurrence or periesophageal regional LN metastasis), 12 (35.3%) were outfield failures, and 6 (17.6%) were both locoregional and outfield failures (Figure 2). The lungs were the most common solid organ affected in cases of outfield failure, accounting for 9 (26.5%) patients. Finally, 22 (43.1%) patients were confirmed dead during the 13-year follow-up period.

### 3.4. Prognostic Significance of Interim Analysis

Factors associated with CR are shown in Table 3. Interim responses were significantly correlated with CR. Reductions to the area and diameter of the primary esophageal lesion in the CR group were significantly higher than those in the non-CR group (18.4% vs. 48.4%, *p* < 0.001, and 16.3% vs. 47.6%, *p* < 0.001, respectively). The reduction of LN area and diameter were also higher in the CR group than in the non-CR group (15.2% vs. 40.6%, *p* = 0.002, and 5.8% vs. 28.7%, *p* < 0.001, respectively). The sum of area and that of diameter reduction were significantly higher in the CR group than in the non-CR group (17.7% vs. 46.5%, *p* < 0.001, and 10.5% versus 40.5%, *p* < 0.001, respectively). However, there was no difference in baseline stages between the CR and non-CR group.

To determine the significance of interim response to treatment failure patterns, we analyzed data from 16 (47.1%) and 12 (35.3%) patients with locoregional and outfield failures, respectively, while excluding 6 (17.6%) patients with both locoregional and outfield failures from the total of 34 patients who experienced treatment failure (Figure 2). There was no difference in the reduction of primary lesion (area and diameter) between the patients with locoregional and those with outfield failures. When the patients experiencing treatment failure were grouped by the degree of interim reduction in the area of the primary esophageal lesion from 5% to 30% in 5%-increments, the proportion of cases with locoregional failure began decreasing at interim response of 20% or higher (*p* = 0.0501), while the proportion of cases with outfield failure followed the opposite pattern, increasing at interim response of 20% or higher (*p* = 0.0533) (Figure 3).

Figure 4 presents Kaplan–Meier curves for OS, depending on interim responses. Area and diameter reduction of the primary esophageal lesion of >30% at interim assessment was associated with increased OS; in contrast, the area reduction of LN was not significantly associated with OS (area: *p*-value 0.001, diameter: *p*-value 0.005) (Figure 4a,b). In cases where the sum of primary esophageal cancer lesion and LN decreased by more than 30%, both area and diameter showed increased survival rate (area: *p* < 0.001, diameter: *p* = 0.014) (Figure 4c). PFS was significantly longer in the patients with ≥30% interim reduction in the area of the primary lesion (*p*-value 0.012) (Appendix A). Table 4 showed the multivariate analyses about CR and OS. Area reduction of the primary esophageal lesion and the sum of area reduction were significantly associated with CR and OS

## 4. Discussion

In the present study, using adaptive CT images, we investigated the role of interim response in predicting survival and overall treatment response among LAESCC patients treated with dCRT. The present study showed that a favorable interim response to dCRT was associated with overall CR rate. The CR group showed significantly higher interim response than did the non-CR group, while the initial lesion area and diameter estimates were comparable between the groups. In addition, the interim response of the primary lesion was associated with OS estimates. A favorable interim response of the LNs tended to be associated with longer OS. However, this association was not statistically significant.

Previously, dysphagia [13], hoarseness, advanced clinical stage [14], and gene expression including YAP1 and Bax [15,16,17,18,19,20,21,22,23] have been associated with poor prognosis in LAESCC. A multicenter study of 181 patients with esophageal cancer treated with dCRT revealed that PET response criteria in solid tumors can help evaluate therapeutic response and the risk of disease progression and death [24,25,26,27,28,29,30]. However, previous studies on early evaluation of dCRT response are rare. In addition, sometimes, it is difficult to evaluate the SUV value of the regional lymph node differentiated from the primary esophageal lesion. The present study has shown the benefits of an interim response evaluation in prognostication after dCRT. Adaptive CT can give the information about tumor burden, both primary esophageal lesion and LN. The use of adaptive CT scanning for interim analysis is both feasible and cost-effective, since ART is part of routine RT protocol at many institutions, helping to deliver accurate and precise dCRT doses [11].

According to multivariate analysis, the response of the primary cancer lesion showed the best correlation with CR and OS estimates. One of the strengths of this study is the investigation of the primary lesion response to treatment. Rather than being a two-dimensional lesion, primary esophageal cancer is a column-shaped, three-dimensional lesion that is challenging to measure and quantify. As a result, treatment response is difficult to assess based on the RECIST criteria, which are otherwise widely used in the evaluation of solid tumors. In the present study, we measured both tumor area and maximum diameter in axial CT images, which allowed us to capture changes to the tumor burden. We emphasized the evaluation of changes to the primary esophageal tumor burden that emerged during dCRT and found a relationship between these changes and the overall treatment response and prognosis.

In the present study, the degree of interim response of the primary esophageal lesion was associated with the type of treatment failure. Specifically, we have demonstrated that the interim primary lesion area reduction during dCRT of >15% corresponded to a higher proportion of cases with outfield treatment failure; meanwhile, the interim reduction of <15 corresponded to a higher proportion of cases with locoregional failure. This information may be helpful to choose treatment strategies such as intensive local treatment or systemic treatment. A large scale study with a long follow-up period may be required to show a significant correlation between the interim response and treatment failure patterns, helping physicians decide treatment strategy.

The present study has some limitations. First, this was a retrospective study, involving a relatively small number of patients. Second, only patients with adaptive CT images taken during dCRT were enrolled; as a result, the study may have been subject to selection bias, as patients with a large tumor are likely to undergo ART during dCRT. Third, the dose range for adaptive RT in the current study is quite large and a higher RT dose at adaptive CT scan is likely to be associated with increased response at the interim analysis. During the time period when the current study cohort underwent dCRT, patients underwent adaptive CT scans based on the initial tumor size (i.e., earlier adaptive CT scans for larger gross tumors) and symptomatic changes (i.e., improvement in swallowing difficulty). Based on the current study results, we now carry out adaptive CT scans at regular intervals of 30.6~41.4 Gy. Lastly, while ROIs were compared at the same level of axial CT image, it is difficult to confirm that the measured areas represented the entire tumor burden at the time of image acquisition. Despite these limitations, this study presents novel evidence of the predictive role of adaptive CT images, which may inform treatment strategies, including dose escalation during dCRT.

## 5. Conclusions

Interim response evaluation using adaptive CT images during dCRT may provide information regarding treatment response and prognosis among patients with LAESCC. Further studies using qualitative evaluation of ROIs, such as texture analysis in a larger population are warranted.

## Figures and Tables

**Figure 1 cancers-13-01255-f001:**
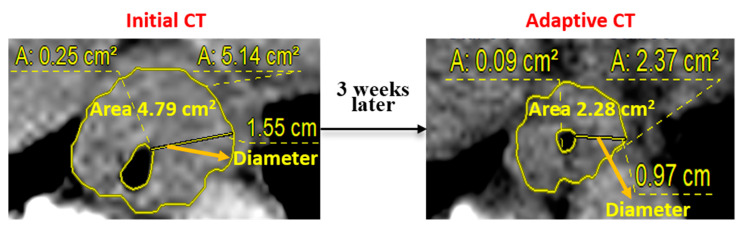
Measurement of region of interest (ROIs) based on initial computed tomography (CT) and adaptive CT scans. The area and maximal diameter of the esophageal lesion in the initial CT scan was 4.79 cm^2^ and 1.55 cm, respectively. In the adaptive CT scan, the corresponding values were 2.28 cm^2^ and 0.97 cm, respectively.

**Figure 2 cancers-13-01255-f002:**
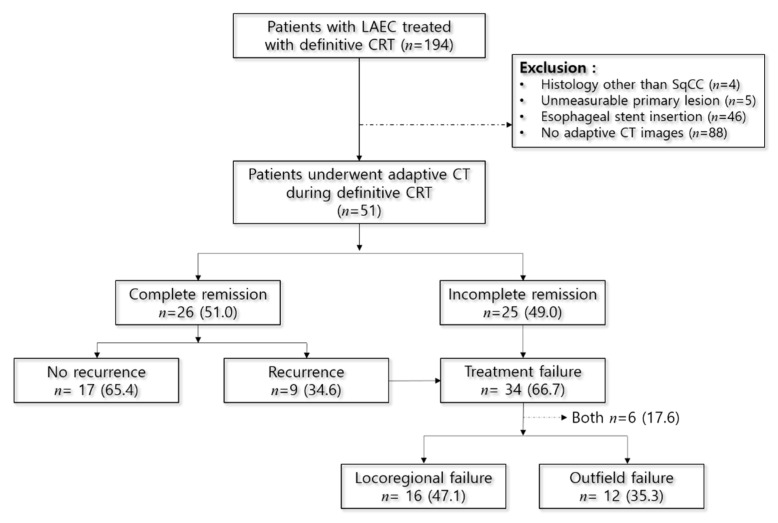
Flow chart capturing treatment outcomes according to the clinical course. LAEC, locally advanced esophageal cancer; CRT, chemoradiotheray; SqCC, squamous cell carcinoma.

**Figure 3 cancers-13-01255-f003:**
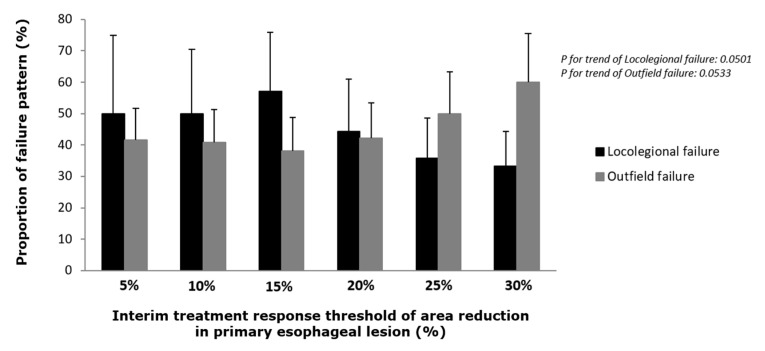
Treatment failure pattern according to the interim response during definitive chemoradiotherapy. The proportion of y-axis refers to the percentage of each treatment failure pattern when the treatment response of the x-axis is higher than that of the *x*-axis value. When the response to treatment estimate was divided by 5%, the rate of local failure tended to increase by up to 15% and turned to be decreased over 15%. Estimates of the outfield failure rate tended to display a pattern in the opposite direction.

**Figure 4 cancers-13-01255-f004:**
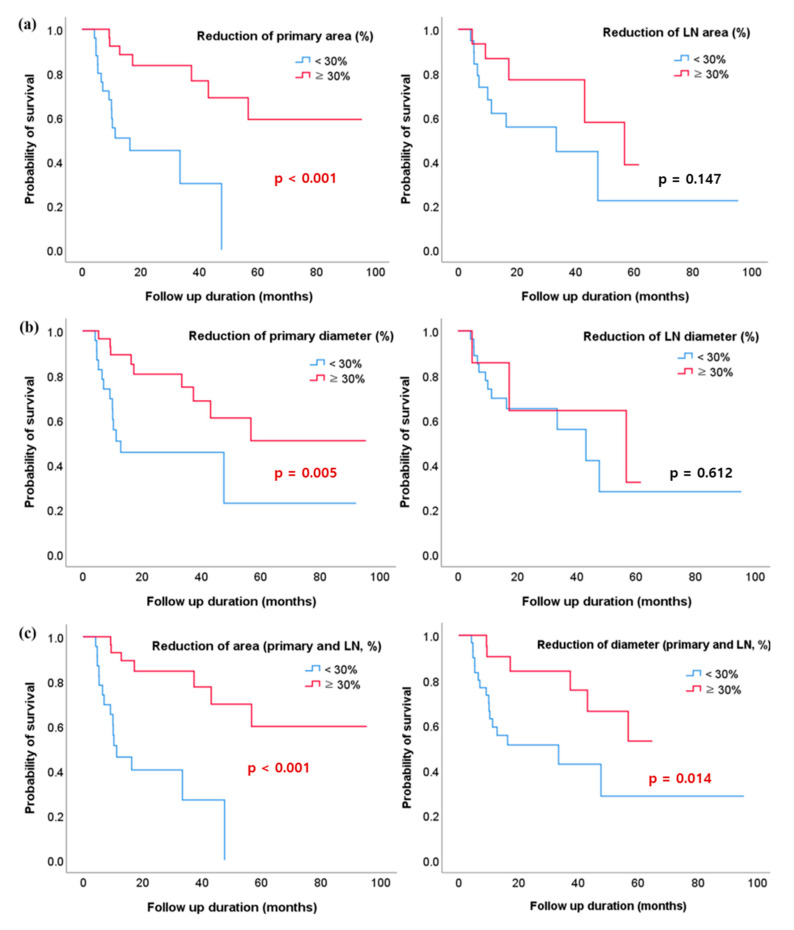
Kaplan–Meier analysis of overall survival (OS) rates shows a relationship between the reduction of lesion size and OS (**a**) Area reduction of the primary esophageal lesion/lymph node (LN) and OS rates; (**b**) Diameter reduction of the primary esophageal lesion/LN and OS rates; (**c**) Area and diameter reduction of both lesions (sum of primary esophageal lesion and LN) and OS rates.

**Table 1 cancers-13-01255-t001:** Baseline characteristics of the patients with esophageal squamous cell carcinoma receiving definitive chemoradiotherapy.

Characteristics	Values (*n* = 51)
Age (years, mean ± SD)	65.6 ± 9.1
Male (n, %)	43 (84.3)
Follow up duration (months) (median, IQR)	16.6 (10.2–41.3) ^a^
Location (n, %)	
Cervical	9 (17.6)
Upper	14 (27.5)
Middle	19 (37.3)
Lower	9 (17.6)
Clinical T Stage (n, %)	
Tx ^b^	2 (3.9)
T1	3 (5.9)
T2	10 (19.6)
T3	27 (53.0)
T4	9 (17.6)
Clinical N Stage (n, %)	
N0	4 (7.8)
N1	23 (45.1)
N2	16 (31.4)
N3	8 (15.7)
Clinical M Stage, n (%)	
M0	51 (100.0)
Clinical TNM stage (n, %)	
Stage II	11 (21.6)
Stage III	31 (60.8)
Stage IVA	9 (17.6)
Tumor histology (n, %)	
WD	7 (13.7)
MD	29 (56.9)
PD	7 (13.7)
N/A (Uncertain invasiveness)	8 (15.7)
Chemotherapy regimen	
5-FU + cisplatin	45 (88.2)
Others ^c^	6 (11.8)
Consolidation chemotherapy (n, %)	27 (52.9)
Adaptive RT dose, cGy [median (min–max)	3570 (2300–6300)
Total dose, cGy [median (min–max)]	6300 (4200–7200)

SD, standard deviation; IQR, interquartile range; AJCC, American Joint Committee on Cancer; WD, well differentiated; MD, moderately differentiated; PD, poorly differentiated; RT, radiotherapy. ^a^ Follow up duration for survivors was 26.2 (15.8–51.2) months. ^b^ Unevaluable was due to the fact that EUS cannot be performed because scope passing is limited. ^c^ 5-FU + carboplatin, Docetaxel + 5-FU + cisplatin, paclitaxel + carboplatin, cisplatin monotherapy.

**Table 2 cancers-13-01255-t002:** Interim treatment response from the comparison between initial and adaptive CT images.

Interim Treatment Response	Values (*n* = 51)
Interval between initial and adaptive CT, days (mean ± SD)	30.6 ± 7.3
Primary esophageal lesion	
Initial area (cm^2^), (median, IQR)	5.9 (3.8–8.6)
Follow up area (cm^2^), (median, IQR)	4.0 (2.3–5.6)
Initial diameter (cm), (median, IQR)	1.8 (1.4–2.7)
Follow up diameter (cm), (median, IQR)	1.3 (0.8–1.6)
Reduction of area (%, mean ± SD)	33.7 ± 24.0
Reduction of diameter (%, mean ± SD)	32.0 ± 23.4
LN	
Initial area (cm^2^), (median, IQR)	2.5 (1.5–3.8)
Follow up area (cm^2^), (median, IQR)	1.7 (1.1–3.3)
Initial diameter (cm), (median, IQR)	2.1 (1.7–2.5)
Follow up diameter (cm), (median, IQR)	1.7 (1.4–2.1)
Reduction of area (%, mean ± SD)	27.2 ± 25.1
Reduction of diameter (%, mean ± SD)	16.6 ± 19.2
Sum of primary lesion and LN	
Initial area (cm^2^), (median, IQR)	8.2 (5.4–12.1)
Follow up area (cm^2^), (median, IQR)	5.7 (3.6–8.1)
Initial diameter, (median, IQR)	3.3 (2.1–4.8)
Follow up diameter, (median, IQR)	2.8 (1.3–3.6)
Reduction of area (%, mean ± SD)	32.4 ± 22.4
Reduction of diameter (%, mean ± SD)	25.8 ± 22.6

SD, standard deviation; IQR, interquartile range; LN, lymph node; CT, computed tomography.

**Table 3 cancers-13-01255-t003:** Factors associated with complete response (CR).

Variables	Non-CR (*n* = 25)	CR (*n* = 26)	*p*-Value
Age (years, mean ± SD)	66.4 ± 8.0	64.9 ± 10.1	0.567
Primary esophageal lesion			
Initial area (cm^2^), (median, IQR)	6.1 (4.1–9.0)	5.7 (2.8–8.5)	0.666
Reduction of area (%, mean ± SD)	18.4 ± 18.0	48.4 ± 19.5	**<0.001**
Initial diameter (cm), (median, IQR)	1.8 (1.4–2.6)	1.8 (1.2–2.7)	0.926
Reduction of diameter (%, mean ± SD)	16.3 ± 20.4	47.6 ± 14.5	**<0.001**
LN			
LN initial area (cm^2^), (median, IQR)	2.5 (1.5–4.1)	2.5 (1.9–3.7)	0.769
Reduction of LN area (%, mean ± SD))	15.2 ± 21.4	40.6 ± 22.4	0.002
LN initial diameter (cm), (median, IQR)	2.0 (1.5–2.6)	2.1 (1.8–2.6)	0.437
Reduction of LN diameter (%, mean ± SD)	5.8 ± 14.6	28.7 ± 16.5	**<0.001**
Sum of primary lesion and LN			
Initial Area (cm^2^), (median, IQR)	8.6 (5.9–12.0)	6.9 (3.6–12.6)	0.254
Reduction of area, sum (%, mean ± SD))	17.7 ± 17.1	46.5 ± 17.5	**<0.001**
Initial diameter (cm), (median, IQR)	3.3 (2.6–4.4)	3.3 (1.9–4.8)	0.618
Reduction of diameter, sum (%, mean ± SD)	10.5 ± 19.3	40.5 ± 14.4	**<0.001**
Clinical Stage (n, %)			0.145
Stage II	4 (16.0)	7 (26.9)	
Stage III	14 (56.0)	17 (65.4)	
Stage IVA	7 (28.0)	2 (7.7)	

SD, standard deviation; IQR, interquartile range; CR, complete response; LN, lymph node. Bold value means statistical significance because *p* value is < 0.05.

**Table 4 cancers-13-01255-t004:** Multivariate analyses about complete response and overall survival rates.

For Complete Response	Univariable Model		Multivariable Model 1		Multivariable Model 2	
	OR(95% CI)	*p*-Value	OR(95% CI)	*p*-Value	OR(95% CI)	*p*-Value
Age	1.02(0.96–1.08)	0.559				
Primary esophageal lesion						
Reduction of area	0.90(0.84–0.95)	<0.001	0.88(0.78–0.99)	0.029		
Reduction of diameter	0.91(0.87–0.96)	<0.001				
Lymph node (LN)						
Reduction of area	0.95(0.91–0.99)	0.008	0.97(0.92–1.03)	0.304		
Reduction of diameter	0.91(0.85–0.97)	0.004				
Sum of primary lesion and LN						
Reduction of area, sum	0.90(0.85–0.95)	<0.001			0.89(0.84–0.95)	<0.001
Reduction of diameter, sum	0.88(0.81–0.94)	<0.001				
Clinical Stage (ref: Stage II)						
Stage III	1.44(0.35–5.95)	0.386	2.79(0.20–39.80)	0.962	2.42(0.38–15.25)	0.764
Stage IVA	6.12(0.83–45.02)	0.064	8.79(0.14–544.29)	0.374	9.62(0.71–130.07)	0.118
For Overall Survival Rates	Univariable Model		Multivariable Model 1		Multivariable Model 2	
	HR(95% CI)	*p*-Value	HR(95% CI)	*p*-Value	HR(95% CI)	*p*-Value
Age	0.99(0.94–1.03)	0.54				
Primary esophageal lesion						
Reduction of area	0.98(0.97–0.99)	0.001	0.98(0.96–0.99)	0.002		
Reduction of diameter	0.97(0.95–0.99)	<0.001				
Lymph node (LN)						
Reduction of LN area	0.99(0.96–1.01)	0.253				
Reduction of LN diameter	0.97(0.94–1.00)	0.07				
Sum of primary lesion and LN						
Reduction of area, sum	0.97(0.96–0.99)	<0.001			0.97(0.95–0.99)	0.001
Reduction of diameter, sum	0.97(0.96–0.99)	<0.001				
Clinical Stage (ref: Stage II)						
Stage III	0.75(0.25–2.19)	0.594	0.58(0.19–1.79)	0.346	0.65(0.22–1.98)	0.454
Stage IVA	2.29(0.72–7.30)	0.16	1.70(0.53–5.50)	0.373	1.84(0.57–5.90)	0.308

## Data Availability

The data presented in this study are available on request from the corresponding author. The data are not publicly available due to privacy and ethical restrictions.

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
