# Peer review of "Prognostic Significance of Interim Response Evaluation during Definitive Chemoradiotherapy for Locally Advanced Esophageal Squamous Cell Carcinoma"

_cancers, 2021, doi:10.3390/cancers13061255_

Round 1
Reviewer 1 Report
- In the method section, authors described that in these axial CT images, regions of interest (ROIs) were manually delineated along the boundaries of tumor lesions. Do the tumor lesions authors delineated include adjacent normal esophagus? Please clarify.
- In the method section, authors described that the maximal diameter of esophageal cancer lesion and diameter of LN were also measured. Changes in these measurements between initial CT and adaptive CT images were calculated. However, in table 1, authors showed there were 4 patients with N0 disease. How did authors measure the diameter of LN in these 4 patients? In table 1, authors also showed there were 3 patients with T1 disease. It is well known that it is difficult to see T1 lesion in CT scan. Usually, T1 lesion is too small to see on CT scan. How did authors delineate the ROIs and measure the diameter of esophageal cancer lesion in these 3 patients with T1 disease?
- In the study, cisplatin and 5-FU are most commonly used regimen. What are the other chemotherapy regimens, paclitaxel/cisplatin or folfox? How many patients received cisplatin/5-FU, paclitaxel/cisplatin or folfox?
- In the result section, authors described that the median total radiation dose was 6300 (4200–7200) cGy. For patients receiving definite CCRT, radiation dose should be at least 5000-5040cGy according the NCCN guideline. Do some patients plan to receive neoadjuvant CCRT as initial treatment protocol? Please clarify.
- The median follow-up duration was 16.6 (10.2–41.3) months. It seems short. How about the median follow-up duration for survivors?
- In table 1, adaptive RT dose was 3570cGy (2300-6300). The dose range 2300-6300 is so large. It means that use of adaptive CT scanning for interim analysis was done at different time point with different radiotherapy dose and the radiation dose range is so large. It is well known that higher radiation dose leads to better response. If interim analyses were done at different radiation dose, did authors find better response under higher adaptive RT dose? It is better to use similar adaptive RT dose to do interim analyses.
- Authors described that Complete response (CR) was defined as the absence of residual tumor confirmed on endoscopic biopsy via histologic examination and a CT or PET scan. What are the criteria to define the absence of residual tumor on CT scan? What are the criteria to define the absence of residual tumor on PET/CT scan?
- How many patients receive salvage esophagectomy in this study?
- Overall survival could be affected by 2nd-line or 3rd-line treatment such as immunotherapy. Could authors showed the impact of interim analyses on progression-free survival?
- For factors associated with complete response (CR) and survival analyses, other clinicopathological factors such as T stage, N stage, AJCC staging, tumor grade …etc should be showed. Besides, we only see univariate survival analysis, how about multivariate analyses? Is interim response an independent prognosticator?
- To determine the significance of interim response to treatment failure patterns, authors analyzed data from 16 (47.1%) and 12 (35.3%) patients with locoregional and outfield failures, respectively, while excluding 6 (17.6%) patients with both locoregional and outfield failures from the total of 34 patients who experienced treatment failure. Authors found the proportion of cases with outfield failure followed the opposite pattern, increasing at interim response of 20% or higher (p = 0.0533). I think this finding has a bias. Authors excluded patients with both locoregional and outfield failures. Patients with outfield failures analyzed by authors are patients with good response in locoregional where interim analyses were done. Hence, this finding is due to exclusion of patients with both locoregional and outfield failures. I suggest that author should not exclude patients with both locoregional and outfield failures. They should be also put in the analysis.
- Many studies demonstrated that pre-CCRT PET or post-CCRT PET can be used to predict CCRT outcome. How about the predict value of PET and the relationship between interim response with pre-CCRT PET or post-CCRT PET?
- Interim PET analysis has been reported to predict CCRT outcome. What are the pros and cons of interim PET and interim CT studied by authors? It is better if authors can discuss it
Author Response
We attached the file about our reply to the review report (Reviewer 1)

Reviewer 2 Report
The prognostic significance of interim response evaluation during dCRT for locally advanced esophageal SCC by Yeom et al is an interesting clinical issue, is only scarcely described in literature and might lead to a clinical decision making.
Remarks.
Introduction: Line 55: the authors describe "potentially resectable (T4a)". I suppose "T2 up to T4a" is ment here.
Methods: Complete response was defined as the absence of residual tumor confirmed on biopsy and CT or PET scan. It is well known that after dCRT for esophageal tumors, mostly a thickening of the esophageal wall remains, and thus cannot be defined as a CR on CT. How did the authors define response when a biopsy was negative with residual disease on CT scan? Did the authors mean that a CR was defined as "either" a negative biopsy or CR on CT scan?
Results:
Only 51 of 194 screened patients were included, most of them excluded because of stent or absence of adaptive scans. This suggests a bias, as also stated by the authors in their discussion. It is likely that mainly measurable bulky tumors are selected for adaptive scans. This is important for interpreting the results and should be noted in more detail in the methods section.
3.3 line 169: treatment failure were reported in 34, which is noted as “35.6%”. On a total of 51 patients this percentage seems incorrect. See also fig 2, where the rate of overall treatment failures is unclear to me.
Fig 4 c, reduction of diameter (primary and LN). In such a small group of patients with overlapping curves, it is hard for me to believe that this difference is significant (p stated as 0.014).
Discussion
The second paragraph is mainly repeating the methods section and does to my opinion not much to the content. I can be shortened or left out.
In the third paragraph an association is made with dose escalation. I do not understand what early response selection has to do with dose escalation, and is not the topic of this study. The referred study no 18 on dose escalation did not select for responding tumors. Furthermore, two randomized studies did not show an effect of dose escalation.
Author Response
We attached the file about our reply to the review report (Reviewer 2)

Round 2
Reviewer 1 Report
Authors answer my questions. I have no further questions.
Reviewer 2 Report
The manuscript has improved, and sufficient answers have been given